# Clinical Outcomes of Individuals with COVID-19 and Tuberculosis during the Pre-Vaccination Period of the Pandemic: A Systematic Review

**DOI:** 10.3390/jcm11195656

**Published:** 2022-09-26

**Authors:** Tulip A. Jhaveri, Celia Fung, Allison N. LaHood, Andrew Lindeborg, Chengbo Zeng, Rifat Rahman, Paul A. Bain, Gustavo E. Velásquez, Carole D. Mitnick

**Affiliations:** 1Division of Infectious Diseases, Department of Medicine, University of Mississippi Medical Center, Jackson, MS 39216, USA; 2Division of Medical Microbiology, Department of Pathology, Brigham and Women’s Hospital, Harvard Medical School, Boston, MA 02115, USA; 3Department of Neurology, Yale New Haven Hospital, New Haven, CT 06510, USA; 4Department of Global Health and Social Medicine, Harvard Medical School, Boston, MA 02115, USA; 5Francis A. Countway Library of Medicine, Harvard Medical School, Boston, MA 02115, USA; 6UCSF Center for Tuberculosis, University of California San Francisco, San Francisco, CA 94110, USA; 7Division of HIV, Infectious Diseases, and Global Medicine, University of California San Francisco, San Francisco, CA 94110, USA; 8Division of Global Health Equity, Department of Medicine, Brigham and Women’s Hospital, Harvard Medical School, Boston, MA 02115, USA

**Keywords:** COVID-19, SARS-CoV-2, tuberculosis, comorbidity, mortality, systematic review, pre-vaccination

## Abstract

Background: Tuberculosis, like COVID-19, is most often a pulmonary disease. The COVID-19 pandemic has severely disrupted tuberculosis services in myriad ways: health facility closures, lockdowns, travel bans, overwhelmed healthcare systems, restricted export of antituberculous drugs, etc. The effects of the shared risk on outcomes of the two diseases is not known, particularly for the first year of the pandemic, during the period before COVID-19 vaccines became widely available. Objective: We embarked on a systematic review to elucidate the consequences of tuberculosis on COVID-19 outcomes and of COVID-19 on tuberculosis outcomes during the pre-vaccination period of the pandemic. Methods: The systematic review protocol is registered in PROSPERO. We conducted an initial search of PubMed, Embase, Web of Science, WHO coronavirus database, medRxiv, bioRxiv, preprints.org, and Google Scholar using terms relating to COVID-19 and tuberculosis. We selected cohort and case–control studies for extraction and assessed quality with the Newcastle-Ottawa scale. Results and Conclusion: We identified 2108 unique abstracts published between December 2019 and January 2021. We extracted data from 18 studies from 8 countries. A total of 650,317 persons had a diagnosis of COVID-19, and 4179 had a diagnosis of current or prior tuberculosis. We explored links between tuberculosis and COVID-19 incidence, mortality, and other adverse outcomes. Nine studies reported on mortality and 13 on other adverse outcomes; results on the association between tuberculosis and COVID-19 mortality/adverse outcomes were heterogenous. Tuberculosis outcomes were not fully available in any studies, due to short follow-up (maximum of 3 months after COVID-19 diagnosis), so the effects of COVID-19 on tuberculosis outcomes could not be assessed. Much of the rapid influx of literature on tuberculosis and COVID-19 during this period was published on preprint servers, and therefore not peer-reviewed. It offered limited examination of the effect of tuberculosis on COVID-19 outcomes and even less on the effect of COVID-19 on tuberculosis treatment outcomes.

## 1. Introduction

Tuberculosis (TB), like coronavirus disease 2019 (COVID-19), is an airborne infectious disease, commonly affecting the lungs. It rages in settings of crowding, poverty, and in individuals with comorbidities including malnutrition, immunosuppression, diabetes mellitus, and substance use disorders [1,2,3,4]. A substantial geographical overlap exists between the two diseases, with high-burden TB countries also listed among top countries for COVID-19 incidence and COVID-19 mortality [5,6]. At the start of COVID-19 pandemic, the extent and direction of the relationship between TB and COVID-19 was unknown. Early analyses predicted that COVID-19—and response measures—would have devastating population effects on TB incidence and mortality [7,8,9,10]. The individual impact of the shared pulmonary risk, and how best to manage it, was unknown. Initial literature searches by our group in March and April 2020 revealed no peer-reviewed publications that directly and systematically examined the individual level effect of COVID-19 disease on TB outcomes and TB disease on COVID-19 outcomes.

To address this gap and systematically assess the relationship between COVID-19 and TB, we embarked on a systematic review. Given the rapidly evolving situation and growing commitment to efficient and open reporting of research during the first year of the pandemic, we expected large numbers of exploratory analyses and frequent publication of results without peer review. Therefore, our systematic review applied very broad inclusion criteria to include such exploratory analyses, preprints, and a wide range of study designs. We aimed to elucidate both the impact of TB on COVID-19 outcomes and the impact of COVID-19 on TB outcomes. We focused on the period of the pandemic before COVID-19 vaccines became widely available. U.S. Food and Drug Administration Emergency Use Authorization approval of the Pfizer-BioNTech COVID-19 Vaccine occurred on 11 December 2020 [11]. Approvals of other vaccines closely followed. However, low- and middle-income countries (LMICs) with high burdens of TB experienced long delays in vaccine access compared to high-income countries [12]. We chose to examine the pre-vaccination period to avoid complicating the interpretation of our results with inequitable access to vaccines as a factor, especially among LMICs where the association between TB and COVID-19 may be most relevant. The objective of this systematic review, therefore, was to answer the two following review questions during the pre-vaccine period: [1] was active or recent TB disease associated with worsened COVID-19 outcomes, and [2] were TB outcomes worsened in the presence of COVID-19?

## 2. Materials and Methods

This systematic review protocol was registered with PROSPERO (CRD42020187349). We conducted two rounds of extraction and review. The initial search was conducted from December 2019 to July 2020; included results were too limited to warrant a publication. A second search was conducted from July 2020 to January 2021. We reported results following Preferred Reporting Items of Systematic Reviews and Meta-Analyses (PRISMA) guidance.

### 2.1. Eligibility Criteria

We conducted searches in PubMed (NCBI), Embase (Elsevier), Web of Science (Clarivate), WHO Global Literature on Coronavirus Disease, medRxiv, bioRxiv, Preprints.org, and Google Scholar using a focused search strategy pertaining to COVID-19 and TB (Appendix A). Languages were limited to English, Spanish, French, and Chinese. Cohort studies and case–control studies were included for data extraction. Case-series, case reports, cross-sectional studies, and ecological studies were considered as Appendix A. Opinion pieces, commentaries, and reviews were excluded. For review question 1, studies were included if they examined the association between TB as exposure (represented as current disease, prior disease and/or infection) and adverse outcomes or severity of presentation of COVID-19 disease. For review question 2, studies were included if they examined the association between COVID-19 as exposure and adverse outcomes of TB disease.

### 2.2. Study Selection and Data Extraction

The articles resulting from the search strategy were uploaded into Covidence (Veritas Health Innovation Ltd., Melbourne, Australia), an online systematic review management software. Each title/abstract was independently reviewed by two reviewers. If the title or abstract mentioned COVID-19 and any TB-related terms, such as for example TB disease, lung disease, BCG, or chest x-ray, the article was accepted for full-text review. Studies that did not have an abstract were added to the full-text review. The remaining studies were independently evaluated by two reviewers to determine if they met eligibility criteria for exclusion.

Data extraction was performed independently by two reviewers. Relevant data was extracted into a standardized Microsoft Excel template. Extracted data include the following: author, title, year of publication, country, setting, description of sample population (demographic and clinical characteristics such as age, sex, tobacco use), study design, exposures (including covariates), outcomes, and crude and adjusted effect estimates. Disagreements between reviewers at the study selection or the full-text review stage were resolved by discussion between the two reviewers, and if necessary, involved consultation with a third reviewer.

For review question 1, we extracted all reported COVID-19 adverse outcomes and grouped them into the following categories: severity of presentation (hypoxia, end organ-failure, stroke), hospitalization, mortality, drug-related adverse events, and non-clearance of virus. For review question 2, we grouped TB outcomes into the following categories: TB treatment interruption/loss to follow up, TB treatment failure, death, hospitalization, sputum smear/culture conversion. For both review questions, we considered the following effect estimates: hazard ratios, risk ratios, odds ratios, and risk differences.

### 2.3. Synthesis of Results

We grouped studies by review question and summarized key characteristics, relevant exposures, and outcomes from cohort studies. For those studies that did not present effect estimates, we calculated risk ratios and 95% confidence intervals from frequencies presented by the authors. For review question 1, we considered TB infection as the exposure and COVID-19-specific outcomes or severity of presentation as endpoints. For review question 2, we considered COVID-19 as the exposure and TB outcomes as the endpoint. We did not perform a meta-analysis for either question given the paucity of uniform reporting of exposures, outcomes, and effect estimates.

### 2.4. Quality Assessment

We used the Newcastle-Ottawa scale to assess the quality and risk of bias of individual cohort and case–control studies [13]. 

## 3. Results

### 3.1. Study Selection

The database searches yielded 2108 unique records. Of these, 511 were selected for full-text review (Figure 1). After screening for relevance and applying inclusion and exclusion criteria, 18 studies were included for data extraction (Table 1). Of these, 9 were peer-reviewed and 9 were preprints. These studies included data from 8 countries; the majority of studies were from China (7/18, 39%). Of the 18 studies, 16 were cohort studies and 2 were case–control studies. In total, they reported on 650,277 persons who had a confirmed diagnosis of COVID-19. Of these, 4179 had been infected with TB (current or prior history of TB). Only 15 studies specifically compared outcomes for persons with COVID-19 and TB versus those with COVID-19 without TB.

### 3.2. Mortality

Of 18 studies included in this review, 9 reported findings on TB and COVID-19 mortality. Studies classified TB exposure, defined study populations, and estimated effects differently. Among exposures, six studies examined TB a priori [14,15,19,20,21,22], while three included TB among a lengthy list of comorbidities and risk factors [23,24,30]. Three studies compared current versus previous TB as a risk factor for death [14,21,22], while 5 studies included comparisons of current and/or prior TB to absence of TB [14,19,20,23,24]. A study from South Africa reported on differing levels of mortality risk by age or comorbidities of individuals with TB and COVID-19 co-infection [15]. Study populations included both hospitalized [14,20,22] and unhospitalized [14] adults. Some studies were conducted among special populations, such as children, pregnant persons, or persons living with cancer [21,29] In 5 studies, the risk of death was estimated using hazard ratios [14,24] relative risks [19] and odds ratios [20,23]; in 4 studies, effect estimates were adjusted for (or propensity score matched on) confounders including location, age, sex, comorbidities, and/or severity of presentation [14,19,20,24]. In 1 study based in Canada, only an unadjusted estimate was presented [23]. In the remaining 4 studies, no effect estimates were reported [15,21,22,29]. In 2 of these, we calculated crude risk ratios from aggregate data presented [21,22].

Results were heterogeneous: studies from high TB burden settings (South Africa, Philippines, Russia) that reported adjusted effect estimates found that current and/or prior TB was associated with increased mortality in nearly all populations [14,19,20,24]; the effect estimate was generally less than 3. In the public health sector in South Africa, in an analysis adjusted for age and sex, people with current TB had a more than 3-fold increase in risk compared to those without TB (adjusted HR 3.29; 95% confidence interval [CI, 2.21–4.88]) [14]. In 2 South African groups of hospitalized COVID-19 patients with a history of TB, no increased risk was detected (adjusted and crude OR 1.1; 95% CI, 0.9–1.317 and adjusted HR 1.09; 95% CI, 0.72–1.6515). In a Canadian population, TB was not identified as a risk factor for COVID-19 mortality [23]. In studies in which we calculated a risk ratio, we did not detect an association between TB and COVID-19 mortality (Table 2) [21,22].

### 3.3. Other Adverse Outcomes Related to COVID-19

Thirteen studies examined TB as a risk factor for adverse outcomes related to COVID-19 besides mortality. These studies used various definitions of TB for exposure and a range of adverse outcomes. Among exposure categories, 7 studies included persons with prior and current TB disease [18,19,20,21,22,24,28]; 3 studies combined prior and current TB as a single exposure [19,20,28] and 4 studies did not specify between prior or current TB [24,25,29,30]. One study used a positive Interferon Gamma Release Assay (IGRA) test result as the exposure; this included persons with active TB disease, history of TB disease, and apparent latent TB infection [28]. The most common adverse outcomes were: severity of COVID-19 disease (7 studies; classifications differed), hospitalization (5 studies; 4 reported proportion hospitalized, 1 reported length of hospitalization), admission to the intensive care unit (2 studies), and receiving invasive mechanical ventilation (2 studies). The full spectrum of adverse outcomes related to COVID-19 reported is summarized in Table 3.

The estimated effect of TB on adverse outcomes related to COVID-19 was variable. Chen et al. found that the risk of severe COVID-19 disease increased in persons with current, prior, and latent TB when compared to those without TB (crude RR 6.19; 95% CI, 1.43–26.90) [28]. Two studies, one from the Philippines and another from South Africa, found that the risk of hospitalization related to COVID-19 was higher among persons with current and prior TB when compared to those without TB (aOR 1.20; 95% CI, 1.04–1.38; 18 crude OR 8.8; 95% CI, 6.9–11.317). A study from Russia did not observe the same association between TB and hospitalization among persons with COVID-19 (aHR 0.95; 95% CI, 0.80–1.14). In contrast, this study reported a decreased risk of admission to the intensive care unit among those with TB exposure (aOR 0.27; 95% CI, 0.11–0.70) [24]. No association was observed between invasive mechanical ventilation and TB exposure [24].

### 3.4. COVID-19 as a Risk Factor for Adverse TB Treatment Outcomes

Among the observational studies reviewed, none systematically characterized the impact of COVID-19 on TB treatment outcomes. However, the case reports and case series (18 studies, 46 cases) identified in the review did highlight certain themes of interest. First, these studies demonstrated that persons admitted for COVID-19 may be diagnosed with active TB during the hospitalization precipitated by COVID-19 [32,33,34,35,36,37,38,39,40,41,42,43,44]. Final clinical outcomes at the conclusion of TB treatment were generally not reported. Second, corticosteroid therapy was used in some persons with COVID-19 and TB co-infection. Some reports cited a TB-specific indication for corticosteroids, namely central nervous system TB [42,43], while others reported the use of corticosteroids as primary therapy for COVID-19 [33,38,45]. Third, persons with COVID-19 and TB co-infection presented with various comorbidities (HIV, diabetes mellitus, chronic kidney disease) [34,37,44,46] and complications: hepatotoxicity [33,35]; extension of the duration of standard TB treatment to 9 months [32]; drug-resistant TB [34,47,48]; extrapulmonary TB [42,43,48]; and death [33,38,48].

### 3.5. Quality Assessment

The average Newcastle-Ottawa score was 4 (range 0–7) out of a total of 9. These results highlight the variable quality of studies included in the analysis. One strength was the use of mortality, a clear endpoint, in 9 studies. In contrast, other outcomes—and exposures—were less well-specified. Another strength in some studies was the use of data from electronic medical records; this reduces the likelihood of reporting bias given that assignment of exposures and outcomes were independent of study activities. In others, exposures were self-reported or extracted from charts by reviewers unblinded to outcome, resulting in an inability to rule out this risk to quality. Several studies did not control for potential confounders, precluding confidence in estimates of an independent effect of exposures. Few studies specified a design that was consistent with the methods used. Overlapping definitions of exposure and outcome and lack of specificity around timing of assessment of exposure and outcome precluded establishment of the temporal relationship between exposure and outcome. For example, in one study, 44 exposures included a disease severity classification, evaluated among predictors for adverse outcomes, which included “peripheral oxygen saturation (SpO_2_) ≤93%”. This complicates interpretation of the effect of TB on adverse outcomes related to COVID-19.

In 5 studies [15,20,23,24,25], large (country or region-wide) databases were used, likely contributing to generalizability of results. In others, population frames were not described, so representativeness of the study population and risk of selection bias could not be ascertained. In most studies, the follow-up period was too short to permit complete outcome reporting and information on loss to follow-up was often omitted. We have summarized biases and/or methodological issues of all included studies in the Appendix A).

## 4. Discussion

The global COVID-19 death toll has reached over 6 million as of this writing [6]. Important immediate and longer-term impacts of the intersecting TB and COVID-19 pandemics have been documented [49]. We set out to better understand how co-occurring TB and COVID-19 affected individuals during the first year of the pandemic, before COVID-19 vaccines became widely available, to examine: (1) if/how does TB affect COVID-19 outcomes and (2) if/how does COVID-19 affect TB outcomes. Even with fairly liberal inclusion criteria, screening 2108 abstracts yielded only 18–9 peer-reviewed—studies that could be extracted. Even these were limited in their ability to inform improved care and prognosis. All extracted studies addressed question 1 and none addressed question 2.

### 4.1. TB as a Risk Factor for Mortality Related to COVID-19

The publications reviewed from Russia, South Africa, and the Philippines suggest that active, current TB disease is associated with an increased risk of COVID-19 mortality [14,19,20,24]. This was not observed in the one large study from Canada [23]. This may be because of the lower burden of TB in Canada than in these other settings; the observable impact of strains on the health sector imposed by the COVID-19 pandemic is likely greatest on conditions most common in those settings, i.e., TB in high-TB-burden settings, non-communicable disease in Canada [50]. On the other hand, in countries with high burden of TB, management becomes more challenging due to high rates of drug-resistant TB [24] increased number of co-morbidities such as HIV infection [20], and lapse in essential TB services [19,51]. An effect of TB on COVID-19 mortality was not detected in smaller studies; these were likely underpowered for this endpoint, including only 22, 18, and 67 persons with COVID-19 each [21,22,29].

A prior systematic review concluded that no independent association could be established between TB and increased COVID-19 mortality [52]. Ecological studies, with countries as the unit of analysis, have produced contradictory results on this relationship. For example, multivariable analysis in one study revealed TB incidence to be one of the strongest predictors of COVID-19 case-fatality rates (standardized coefficient 3.15; 95% CI, 1.09–5.22; *p* = 0.004) [53]. Three other studies found TB prevalence or TB incidence to be inversely correlated with COVID-19 mortality [54,55]. One found this relationship to be true in high- and low-middle-income countries while no significant relationship was observed in low- and upper-middle-income countries [56].

Several additional factors may affect the ability to ascertain this relationship. Much has been made of the potential for BCG vaccination to protect against COVID-19. Theoretically, BCG vaccine could reduce both the risks of TB and COVID-19 mortality, possibly obscuring an increased risk of COVID-19 mortality among those with TB. However, the limited, variable efficacy of BCG in preventing adult TB is well established [57]; its impact on COVID-19 mortality is, to date, at best equivocal [58,59,60]. In addition, in the studies included in the current review, several methodological issues may compromise accurate, valid estimation of the relationship. First is potential misclassification of TB: most of the studies relied on retrospective record review complicating the ability to ensure that exposure and outcome were specified consistently. While death (the outcome) can be confidently established, prior, treated, current, and prior and current TB were all considered as exposures and definitions were rarely specified. It is not known whether “treatment” referred exclusively to chemotherapy for active disease or might also have included chemoprophylaxis for latent TB infection. If those “treated” included those who received chemoprophylaxis, such misclassification might bias any effect estimate toward the null, obscuring a relationship between active TB disease and COVID-19 mortality. Second, only a minority of studies reported effect estimates adjusted for potential confounders. This precludes drawing inference about the independent association between TB and mortality. Third, selection bias may obscure an association between TB disease and COVID-19 mortality. If, for example, persons with a history of TB were more likely to be hospitalized for COVID-19 than persons with no history of TB, studies of mortality in hospitalized patients would likely overestimate the risk of mortality among hospitalized patients with a history of TB.

The totality of the findings on persons with current TB is suggestive of an increased risk of COVID-19 mortality during the pre-vaccination period. Persons with concomitant TB and COVID-19 should likely receive priority care for COVID-19. While the treatment of COVID-19 with immunosuppressive medications such as glucocorticoids or immunomodulatory medications such as IL-6 antagonists is supported by current data for specific clinical scenarios [61,62]; these data are not necessarily generalizable to persons with concomitant active TB. Thus, the risks and benefits of these medications must be weighed carefully in patients with TB and COVID-19 co-infection, given these drugs’ potential for exacerbating TB disease. Moreover, in the absence of compelling evidence that rules out the possible relationship of prior TB and COVID-19 mortality, patients with a history of active TB disease should be considered to have elevated risk; their vaccination should be prioritized. The documented high prevalence of post-TB pulmonary disease [63,64,65,66] and overrepresentation of marginalized and vulnerable populations among persons with a history of TB further support the idea that they might be at increased risk of COVID-19 mortality [67,68].

### 4.2. TB as a Risk Factor for Adverse Outcomes Related to COVID-19

Thirteen studies evaluated the association between TB and adverse outcomes related to COVID-19 besides mortality during the pre-vaccination period. The most common outcomes were severity of COVID-19 disease, hospitalization, admission to the intensive care unit, and need for invasive mechanical ventilation. The aforementioned inconsistent definitions of TB exposure across studies made it difficult to establish (or rule out) a link between TB and adverse outcomes related to COVID-19.

Specifically, a statistically significant increased risk of severe COVID-19 disease was found for persons with prior or current TB in only one study with a sample of 36 cases among the 7 studies that examined this association [28]. We speculate that (1) this detected association may not be real; instead it is an artifact of some selection in the sample or (2) the differences in exposure classification preclude generalizable associations between TB and COVID-19 outcomes: 2 of the studies only investigated prior TB as a risk factor [26,31], 3 did not distinguish between prior TB and TB concurrent with COVID-19 diagnosis [25,29,30], and 1 examined concurrent TB, differentiating between TB diagnosed during COVID-19 and prior to hospitalization [21]. These differences highlight the challenge of conducting a formal meta-analysis, or drawing even informal inference across studies. These exercises would require re-assignment of the exposure in some of the studies and would likely result in misclassification. For example, if a study were to examine a population of patients with prior TB, researchers could erroneously group that population with COVID-19 patients with current TB. If current TB increases the risk of severity and prior TB does not, grouping these two exposures would underestimate the effect of current TB; a meta-analysis would likely return an effect estimate biased toward the null. Alternatively, if patients with a history of TB were grouped with “no TB” and a history of TB increases the risk of severity of COVID-19, then the effect estimate would also be biased toward the null. A similar issue limits the ability to combine results from studies that sought to analyze the association between TB and admission to the intensive care unit [22,24]. One study found a risk reduction among those with prior or current TB (without distinguishing between these) [24]; the other found no difference when comparing between prior, treated, and current TB [22]. This again illustrates the possibility of biased effect estimates if findings were to be combined across studies.

In addition to varying definitions of TB as an exposure, the definitions of several COVID-19 outcomes were not consistent across the included studies. For example, 7 studies reported on severity of COVID-19 outcomes. Among these, 5 studies were conducted in China, 1 in Qatar and 1 in India. The studies conducted in China defined severity using the standardized COVID-19 severity guidelines provided by the National Health Commission of the People’s Republic of China, while the study in Qatar referenced the WHO interim guidance for definitions of severity, and the study in India did not specify which guidelines were used. Referencing different standardized scales for COVID-19 severity and grouping them for combined outcome analysis may result in misclassification if the raw data on the components of the scales are not consistently available. When reporting the severity of COVID-19 infection among pregnant persons with COVID-19 and current or prior TB in India, Gajbhiye did not define severe disease presentation. Without standardized definitions of mild, moderate, or severe COVID-19 disease, people could be misclassified into the different outcome groups, which may also lead to misleading conclusions.

Finally, the geographical variability among the 13 studies is an additional barrier to drawing meaningful conclusions regarding the effect of TB on adverse outcomes related to COVID-19. The included study countries had vastly different burdens of COVID-19 and TB. For example, India, South Africa, Turkey, China, and Philippines had high burdens of COVID-19 while Canada and Qatar do not [6]. It is reasonable to speculate that policies and the availability of resources influenced likelihood of hospitalization and other healthcare workers’ decision making for clinical management. For example, since India had a high burden of COVID-19 and an overwhelmed healthcare system, hospitalization might have been reserved only for persons who were critically ill [69]. However, in China, all suspected and confirmed cases of COVID-19 were quarantined and treated in designated hospitals. During the outbreaks of COVID-19 in Wuhan, the local government launched 47 COVID-19 designated hospitals (including 2 new hospitals) and constructed one shelter hospital for quarantine and treatment [70]. When considering persons coinfected with TB and COVID-19, countries with a high burden of COVID-19 and limited healthcare resources might not admit these cases to the hospital for treatment in fear that they will expose admitted COVID-19 patients to TB. Ultimately, a formal meta-analysis might yield a biased estimate of the impact of TB on COVID-19 hospitalization.

Three previously published systematic reviews examined the effects of TB and on adverse outcomes related to COVID-19, 2 of which included meta-analysis. All 3 reviews had some overlap in included articles with the present review. One included a meta-analysis of COVID-19 occurrence and severity across studies that classified TB exposure differentially as HIV/TB or TB and did not distinguish between prior and current TB, in analyses of mortality or COVID-19 severity as outcomes [71]. The other examined the effects of COVID-19 on TB management and did not conduct a meta-analysis. The authors only included case reports and case series to describe the clinical characteristics and outcomes of persons coinfected with TB and COVID-19; they could not make generalizations about the relationship of TB on adverse outcomes related to COVID-19 [72]. The third systematic review included a meta-analysis of the proportion of persons with COVID-19 with active pulmonary TB, and the relative risk for reported adverse outcome. The authors concluded that the risk of COVID-19 severity and COVID-19 mortality is increased in patients with active pulmonary TB [73]. Even with the additional literature published subsequent to our review, we determined that performing a meta-analysis would be inappropriate given the described inconsistent definitions of exposure and outcome measures, and varying environments where the studies were conducted.

### 4.3. COVID-19 as a Risk Factor for Adverse TB Treatment Outcomes

For a small number of persons included in the case reports and case series reviewed here, admission for COVID-19 prompted further workup and the establishment of a concomitant, new diagnosis of active TB. While it is possible that COVID-19 requiring acute presentation to care may have facilitated earlier diagnosis of TB in these cases, the more likely scenario is that many more people with TB went undiagnosed or suffered delays to diagnosis during the COVID-19 pandemic, given the overlap of symptomatology and radiographic findings for COVID-19 and active TB [74]. Even without the pandemic, loss at each stage of the cascade of care constitutes a critical problem contributing to poor TB care delivery outcomes [75,76]. the COVID-19 pandemic has likely widened diagnostic and treatment gaps in the TB care cascade.

Evidence for widening gaps at the diagnostic stage of the TB care cascade, related to the COVID-19 pandemic, has emerged from India, Indonesia, and the Philippines. These countries reported a 25–30% reduction in TB diagnoses for the period January–June 2020 compared with the same period in 2019 [77]. Two reports from China demonstrated reductions in the number of new TB diagnoses and treatment completion, and increases in the number of persons with TB who had postponed or missed follow-up examinations [78,79]. A retrospective study from South Africa reported a 21–38% reduction in rifampin-resistant TB diagnoses between March 2020 and February 2021 [80], compared to corresponding quarters from the previous year [53].

Loss from the treatment stage of the care cascade has also been reported. For example, at 2 major health centers in Canada, the adjusted rate ratio for weekly active TB treatment initiations was lower in the COVID-19 era by 16% and 29%, respectively, compared to the pre-COVID-19 era [81]. This worsens an already significant global gap: prior to the COVID-19 pandemic, among the 10 million persons estimated to develop active TB annually, only 70.5% received treatment [77]. Although this systematic review did not conclusively link COVID-19 exposure with adverse TB treatment outcomes on an individual level, the observed impact of the pandemic on population-level TB indicators is substantial. In 2020, there were roughly 1.5 million TB deaths worldwide, representing an increase in TB mortality for the first time since 2005 [74]. World Health Organization modeling signals that the pandemic will contribute to an exacerbation of TB incidence and mortality [82].

Some considerations for clinical management and questions for further research emerge from the limited reports about COVID-19 as risk factor for adverse TB treatment outcomes. A vast majority of these patients received hydroxychloroquine, which is no longer the standard of care for COVID-19 directed treatment and could have possibly contributed to adverse outcomes [83]. A Among 4 persons coinfected with TB and COVID-19 who received corticosteroid therapy for COVID-19, two deaths were reported [33,38,45]. The use of corticosteroids for severe/critically ill, hospitalized persons with COVID-19 [84] raises the question about whether TB outcomes might, in fact, be worsened among persons coinfected with active TB and COVID-19 [38]. The level of evidence reviewed here is insufficient to draw conclusions on either point. Among the 5 persons coinfected with TB, HIV, and COVID-19 reviewed here, all 5 demonstrated clinical improvement with directed therapy for TB and COVID-19 [37,44,46]. Lastly, COVID-19 has been associated with abnormal liver function tests [85]. This may complicate the initiation or tolerance of TB therapy given the known hepatotoxicity of several antituberculous agents, as documented for two persons among the reviewed studies [33,35].

### 4.4. Strengths and Limitations

The major strength of this systematic review is the large sample of studies available for review from the period before vaccination against COVID-19 was widely available; studies included populations across a range of geographic settings; in public- and private-sector health systems, in some cases drawing on national samples. Moreover, examining the pre-vaccination period removed inequitable access to vaccines and the widespread presence of subsequent variants as complicating factors to our review. That half the papers extracted were not yet peer-reviewed is both a strength and a limitation. Investigators worked to quickly share results from their research in the pandemic setting. The value of conducting—and releasing results from—high-quality research as quickly as possible cannot be overstated. Although there are clear advantages to quick publication of findings that inform care during a rapidly evolving and deadly pandemic [86]. some aspects of the work, and its interpretation, may suffer. Peer review, though imperfect, offers an opportunity to identify errors, ensure clarity and completeness, and improve interpretation prior to publication. Some of the challenges that we identified in publications returned in our search, and that led to the low scores on the Newcastle-Ottawa assessment, may emanate, in part, from the lack of peer review. This includes failure to clearly describe the study design and resulting errors in analysis strategies as well as poor specification of the exposure and some of the study endpoints. For studies that did not have well-specified designs, we referenced epidemiologic textbook guidance to inform possible reclassification of the design for this review [87]. These limitations, together with the failure to consistently report adjusted effect estimates, led to an inability to perform meta-analysis and to meaningfully interpret the results across the studies. It also suggests that one must interpret existing meta-analyses of TB and COVID-19 with caution.

It is important to acknowledge the challenge of doing quality research during a pandemic that kills healthcare workers and their patients while devasting underlying health systems. Despite this enormous constraint, precedent reveals that it is not impossible. A systematic review of preprints during the recent Ebola virus disease (EVD) and Zika epidemics reported that 60% of EVD and 48% of Zika preprints were later peer-reviewed and published in PubMed-indexed journals, including in very high-profile journals (e.g., approximately 100 articles in Lancet family journals) [86]. In the case of COVID-19, 40% of early COVID-19 preprints were peer-reviewed and published before October 2020. As research burgeoned in the latter half of the year <10% of preprints went on to peer-review [88]. This underlines the reality that there is a staggering amount of data related to COVID-19 posted to preprint servers that has not been peer-reviewed in a timely manner.

While quality research is possible, even in the midst of a pandemic that affects researchers and participants83, it is worth considering study designs that could produce results that would better address the important questions about TB and COVID-19. For question 1, with a focus on mortality or other adverse outcomes, the papers by Sy et al. and Boulle et al. are strong examples of retrospective designs to estimate the risk of TB on COVID-19 mortality. An improvement could come through assurance of properly classified exposure: to achieve this, all cohort members would be evaluated for TB. A prospective cohort of people screened for TB, where exposed people are diagnosed with TB and unexposed people have TB ruled out, would permit this. Unexposed and exposed would be drawn from the same setting (whether community or health centers or occupational group) and underlying population and ideally would have similar risk profile (age, comorbidities, socioeconomic conditions); they should be eligible to receive same quality/intensity of care (i.e., have the same health insurance status or access to care). Optimally, assignment of well-specified and consistently measured/reported outcomes would be blind to exposure. For review question 2, as noted, there were no publications of cohort studies to answer this question. This is likely because the ideal design would involve a prospective cohort of TB patients (from the same health facilities or catchment areas) followed from TB diagnosis until death or 6 months after completion of TB treatment (whichever comes first). They would be tested (RT-PCR) serially for SARS-CoV-2 at standardized times or followed systematically for a standard set of symptoms. All their time on study during which COVID-19 is ruled out would contribute to unexposed time. All their time on study after COVID-19 is diagnosed would contribute to exposed time. The study would then compare incidence of unfavorable outcomes (e.g., death, treatment failure, recurrent TB, lung function at end of treatment) between exposed and unexposed. Alternatively, a case-control study in TB patients could be organized. Cases would be people who died or experienced treatment failure or recurrent TB, or people who have poor lung function post treatment; controls would be people in whom all these outcomes are assessed and ruled out. One would evaluate the difference in COVID-19 incidence in the two groups, matching on/controlling for likely confounders (e.g., HIV, diabetes, BMI, vaccination status). Exposure misclassification would be a risk in this design if standards for testing for SARS-CoV-2 were not established and followed.

## 5. Conclusions

Despite the above limitations, our systematic review does suggest that outcomes may have been worse in individuals infected with tuberculosis and COVID-19 during the pre-vaccination period, and thus these individuals should be considered at a slightly increased risk for mortality. High COVID-19 mortality is expected in TB patients due to TB program disruptions, diagnostic delays, treatment interruptions, and lack of access to drugs [51]. In addition, health facility closures and reduced ability to pay for medical costs disproportionately affect socially disadvantaged populations, compounding already existing health inequalities, particularly in countries with already vulnerable health systems [89].

This review provides a summary of existing literature for COVID-19 and TB outcomes in the pre-vaccination period. Even with the rapid influx of literature on COVID-19 and TB, little inference can be drawn about the effects of tuberculosis on COVID-19 outcomes and of COVID-19 on TB outcomes. This highlights the critical importance of distribution of sufficient resources to make it possible to care for people with COVID-19, TB, and both as well as to identify important risks and interventions that can modify these risks. Much is made of incompatibility between services for raging, infectious, lethal diseases and research. Lessons learned from other epidemics (HIV, EVD), however, highlight the importance of ethical ways research can be performed while not interfering with care, strengthening health systems, and bolstering future capacity to manage and conduct research in pandemics.

## Figures and Tables

**Figure 1 jcm-11-05656-f001:**
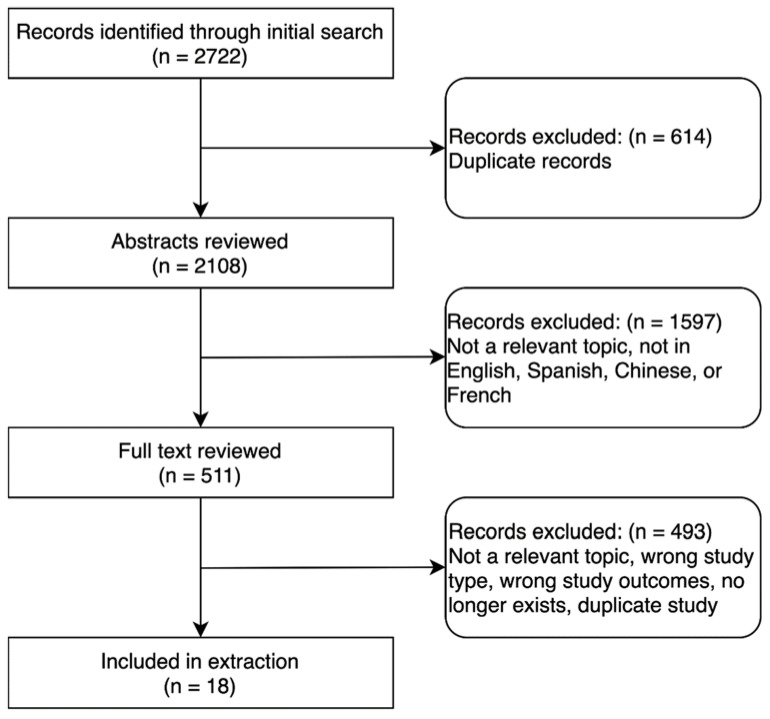
Selection of studies.

**Table 1 jcm-11-05656-t001:** Characteristics of all included studies reporting on COVID-19 endpoints with TB as exposure.

Study No. (Reference No.)	First Author (Publication Date)	Location	Study Population	Sample Size	Study Design	Study Endpoints
1 [14]	Boulle (August 2020)	Western Cape, South Africa	Persons ≥ 20 years old from the public-sector Western Cape Provincial Health Data Center who were not known to have died before 1 March 2020, and with follow-up through 9 June 2020	22,308	Retrospective cohort	COVID-19 mortality
2 [15]	Pillay-van Wyk (October 2020)	South Africa	Persons reported to the National Department of Health as having died from COVID-19, 28 March 2020–3 July 2020	3088	Retrospective cohort	COVID-19 mortality
3 [16]	Torun (October 2020)	Turkey	Healthcare workers with confirmed COVID-19 from 3 hospitals who responded to an online questionnaire	465	Retrospective cohort	COVID-19 hospitalization and radiological pneumonia
4 [17]	Sun (April 2020)	Beijing, China	Persons hospitalized with confirmed, non-imported COVID-19	63	Retrospective cohort	COVID-19 disease severity
5 [18]	van der Zalm (June 2021)	Cape Town, South Africa	Children 0–13 years old with confirmed COVID-19 presenting to Tygerberg Hospital, 17 April 2020–24 July 2020	159	Retrospective cohort	COVID-19 hospitalization
6 [19]	Sy (July 2020)	Philippines	Persons with COVID-19 reported by 17 May 2020 and followed until 17 June 2020	12,513	Prospective cohort	COVID-19 mortality, recovery from COVID-19
7 [20]	Jassat (December 2020)	South Africa	Persons with confirmed COVID-19 reported in national surveillance system (DATCOV) for COVID-19 hospitalization, 5 March 2020–11 August 2020	41,877	Retrospective cohort	COVID-19 related in-hospital mortality
8 [21]	Gajbhiye (February 2021)	Mumbai, India	Pregnant and postpartum women with COVID-19 admitted to BYL Nair Hospital, April 2020–September 2020	18	Retrospective cohort	Maternal outcomes, including COVID-19 mortality
9 [22]	Gupta (October 2020)	New Delhi, India	Persons admitted to Safdarjung Hospital with COVID-19 and current or prior treated TB, 1 February 2020–14 June 2020	22	Retrospective cohort	COVID-19 mortality, ICU, IMV
10 [23]	Fisman (September 2020)	Ontario, Canada	Persons with confirmed COVID-19, 23 January 2020–15 May 2020	21,992	Retrospective cohort	COVID-19 mortality
11 [24]	Demkina (November 2020)	Russia	Persons hospitalized with and treated for COVID-19 reported in the Federal Register of COVID-19 Patients, 26 March 2020–3 June 2020	541,377	Retrospective cohort	Length of COVID-19 hospitalization, mortality, ICU transfer, IMV
12 [25]	Kuwari (July 2020)	Qatar	Persons with confirmed COVID-19 reported to the Ministry of Public Health, 28 February 2020–18 April 2020	5685	Retrospective cohort	COVID-19 disease severity
13 [26]	Meizhu Chen (April 2020)	Zhuhai, China	Persons hospitalized with COVID-19 at Fifth Affiliated Hospital of Sun Yat-sen University, 17 January 2020–10 March 2020	97	Retrospective cohort	COVID-19 disease severity
14 [27]	Lei (March 2020)	Daofu, Sichuan, China	Persons from Tibet hospitalized with confirmed COVID-19 at Daofu People’s Hospital, 4 January 2020–28 February 2020, and followed until 5 March 2020	67	Prospective cohort	COVID-19 symptoms
15 [28]	Yu Chen (March 2020)	Shenyang, China	Persons hospitalized with confirmed COVID-19 at 4 hospitals, 26 January 2020–15 February 2020	36	Case control	COVID-19 disease severity
16 [29]	Hongyan Zhang (September 2020)	Wuhan, Hubei, China	Persons with cancer and COVID-19 treated at 5 hospitals, 5 January 2020–March 18 2020	107	Retrospective cohort	COVID-19 mortality and severity
17 [30]	Huizheng Zhang (March 2020)	Chongqing, China	Persons hospitalized with COVID-19 at Chongqing Public Health Medical Center, 11 February 2020–28 February 2020	43	Retrospective cohort	COVID-19 disease severity
18 [31]	Bi (May 2020)	Shenzhen, China	Persons hospitalized with confirmed COVID-19 at Shenzhen Third People’s Hospital, 11 January 2020–10 March 2020, and followed until 7 April 2020	420	Prospective cohort	COVID-19 disease severity

Abbreviations: ICU, intensive care unit; IMV, invasive mechanical ventilation.

**Table 2 jcm-11-05656-t002:** Effect estimates for the association between TB and mortality related to COVID-19, by study.

Study No. (Reference No.)	Exposure	COVID-19-Related Deaths/ Exposed (%)	COVID-19-Related Deaths/ Unexposed (%)	Unadjusted Estimate (95% CI)	Adjusted Estimate (95% CI)
1 [14]	Boulle, August 2020 (South Africa)				
1a. Population: All persons in the public sector					
	Prior TB	87/1785 (4.9%)	512/20,180 (2.5%)	None	1.79 (1.42–2.24) ^a,h^
				None	1.81 (1.44–2.28) ^a,i^
				None	1.51 (1.18–1.93) ^a,b^
	Current TB	26/343 (7.6%)	512/20,180 (2.5%)	None	2.79 (1.88–4.13) ^a,h^
				None	3.29 (2.21–4.88) ^a,i^
				None	2.70 (1.81–4.04) ^a,b^
1b. Population: Persons in the public sector diagnosed with COVID-19 before 06/01/2020					
	Prior TB	74/1254 (5.9%)	414/13,744 (3.0%)	None	1.55 (1.19–2.02) ^a^
	Current TB	22/235 (9.4%)	414/13,744 (3.0%)	None	1.62 (1.04–2.51) ^a^
1c. Population: Persons hospitalized in the public sector diagnosed with COVID-19 before 06/01/2020					
	Prior TB	77/321 (24.0%)	448/2509 (17.9%)	None	1.40 (1.08–1.82) ^a^
	Current TB	25/148 (16.9%)	448/2509 (17.9%)	None	1.09 (0.72–1.65) ^a^
2 [15]	Pillay-van Wyk, October 2020 (South Africa)				
Population: All persons diagnosed with COVID-19 and current TB reported to the National Department of Health as having died from COVID-19					
	Age		None	None	None
	<50	37/476 (7.8%)			
	50–69	33/1270 (2.6%)			
	>=70	10/704 (1.4%)			
	Sex		None	None	None
	Male	50/1259 (4.0%)			
	Female	30/1198 (2.5%)			
	Location		None	None	None
	Western Cape	62/1587 (3.9%)			
	Eastern Cape	8/406 (2.0%)			
	Gauteng	6/312 (1.9%)			
	Other provinces	4/152 (2.6%)			
6 [19]	Sy, July 2020 (Philippines)				
	Current and prior TB—full cohort	25/106 (23.6%)	46/424 (10.8%)	NA	2.17 (1.40–3.37) ^c,f^
	Current and prior TB—hospitalized patients	18/66 (27.3%)	32/264 (12.1%)	NA	2.25 (1.35–3.75) ^c,f^
7 [20]	Jassat, December 2020 (South Africa)				
	No history of TB		6469/34,464 (18.8%)	Reference	Reference
	Prior TB	202/741 (27.3%)		1.10 (0.90–1.30) ^d^	1.30 ^g^
	Current TB	59/238 (24.8%)		1.60 (1.20–2.20) ^d^	2.00 ^g^
	Current and prior TB	92/346 (26.6%)		1.10 (0.90–1.40) ^d^	2.20 ^g^
8 [21]	Gajbhiye, February 2020 (India)				
	Current TB	1/6 (16.7%)		5.57 (0.26–119.53) ^f^	
	Prior (treated) TB	0/12 (0.0%)		Reference	
9 [22]	Gupta, October 2020 (India)				
	Current TB	3/13 (23.1%)		0.69 (0.18–2.69) ^f^	
	Prior (treated) TB	3/9 (33.3%)		Reference	
10 [23]	Fisman, September 2020 (Canada)				
	TB cases (timing not specified)	NR/52		0.88 (0.21–3.70) ^d^	
11 [24]	Demkina, November 2020 (Russia)				
	TB cases (timing not specified)	NR/324	NR/541,053		1.74 (1.11–2.71) ^a,j^
16 [29]	Hongyan Zhang, March 2020 (China)				
	TB cases (timing not specified)	0/3 (0.0%)	23/104 (22.1%)	None	None

Abbreviations: CI, confidence interval; COVID-19, coronavirus disease 2019; NA, not applicable; NR, not reported; TB, tuberculosis. ^a^ Hazard ratio. ^b^ Adjusted for age, sex, diabetes, hypertension, chronic kidney disease, chronic pulmonary disease/asthma, and infection with human immunodeficiency virus. ^c^ Risk ratio. The risk ratio was calculated using the frequencies provided in the publications. ^d^ Odds ratio. ^e^ Regression coefficient. ^f^ Results are reported on a propensity score matched cohort, matched on: age, sex, chronic obstructive pulmonary disease, asthma, diabetes, hypertension, cancer, renal disease, cardiac disease, and autoimmune disorders. Each person with COVID-19 and TB was matched to four persons with COVID-19 without TB, using nearest neighbor matching of propensity scores, a caliper of 0.05, and without replacement. ^g^ Study did not report the exact 95% CI. ^h^ Adjusted for location. ^i^ Adjusted for age and sex. ^j^ Adjusted for age, sex, influenza vaccination, comorbidities (pulmonary disease, cardiovascular disease, endocrine system disease, cancer/metastasis, infection with human immunodeficiency virus), COVID-19 diagnosis, intensive care unit transfer, invasive mechanical ventilation, disease progression, and oxygen saturation.

**Table 3 jcm-11-05656-t003:** Effect estimates for the association between TB and other adverse outcomes related to COVID-19, by endpoint and study.

Study No. (Reference No.)	Exposure	Events/ Exposed (%)	Events/ Unexposed (%)	Unadjusted Estimate (95% CI)	Adjusted Estimate (95% CI)
Endpoint: Hospitalization					
3 [16]	Torun, October 2020 (Turkey)				
TB treatment history		4/4 (100.0%)	58/132 (43.9%)	None	None
Positive PPD		39/96 (40.6%)	178/369 (48.2%)	0.84 (0.65–1.10) ^a^	None
Radiological pneumonia		116/138 (84.1%)	69/263 (26.2%)	3.20 (2.58–3.97) ^a^	None
5 [18]	van der Zalm, June 2021 (South Africa)				
Children with current pulmonary TB vs. none		2/2 (100.0%)	60/157 (38.2%)	None	None
6 [19]	Sy, July 2020 (Philippines)				
Current and prior TB vs. none ^e^		67/106 (63.2%)	236/424 (55.7%)	None	1.20 (1.04–1.38) ^b^
7 [20]	Jassat, December 2020 (South Africa)				
Prior TB vs. none		341/741 (46%)	2403/34,184 (7.0%)	4.0 (3.3–4.9) ^c^	None
Current TB vs. none		80/238 (33.6%)	2403/34,184 (7.0%)	7.70 (5.40–10.90) ^c^	None
Current and prior TB vs. none		212/346 (61.3%)	2403/34,184 (7.0%)	8.80 (6.90–11.30) ^c^	None
Endpoint: Severity					
8 [21]	Gajbhiye, 2021 (India)				
Current TB (diagnosed in course of COVID-19 hospitalization vs. prior to hospitalization)		1/6 (33.3%)	None	None	None
12 [25]	Kuwari, 2020 (Qatar)				
TB (timing not specified)		3/13 (23.1%)	291/5449 (5.3%)	2.55 (0.94–6.90) ^a^	None
13 [26]	Meizhu Chen, 2020 (China)				
Prior TB vs. none		2/2 (100.0%)	24/95 (25.3%)	None	None
15 [28]	Yu Chen, 2020 (China)				
Current active TB		3/3 (100%)	2/23 (8.7%)	None	None
Current latent TB		0/2 (0.0%)	2/23 (8.7%)	None	None
Prior TB		4/8 (50.0%)	2/23 (8.7%)	5.25 (1.18–23.28) ^a^	None
Current and prior TB		7/13 (77.8%)	2/23 (8/7%)	6.19 (1.43–26.90) ^a^	None
16 [29]	Hongyan Zhang, 2020 (China)				
TB (timing not specified) vs. none		0/3 (0.0%)	56/104 (53.9%)	None	None
17 [30]	Huizheng Zhang, 2020 (China)				
TB (timing not specified) vs. none		0/1 (0.0%)	14/42 (33.3%)	None	None
18 [31]	Bi, 2020 (China)				
	Prior TB vs. none	1/9 (11.11%)	92/411 (22.4%)	0.50 (0.08–3.20) ^a^	None
Endpoint: Admitted to the intensive care unit					
9 [22]	Gupta, 2020 (India)				
Prior (treated) vs. current active TB patients		4/13 (30.8%)	3/9 (33.3%)	0.92 (0.27–3.17) ^a^	None
11 [24]	Demkina, November, 2020 (Russia)				
TB (timing not specified)		Not reported	Not reported	Not reported	0.27 (0.11–0.70) ^c^
Endpoint: Received invasive mechanical ventilation					
9 [22]	Gupta, 2020 (India)				
Prior (treated) vs. current active TB patients		3/13 (23.1%)	3/9 (33.3%)	0.69 (0.17–2.69) ^a^	None
11 [24]	Demkina, November, 2020 (Russia)				
TB (timing not specified)		Not reported	Not reported	0.80 (0.33–1.94)	None
Endpoint: Radiologic pneumonia					
3 [16]	Torun, October 2020 (Turkey)				
TB treatment history		3/4 (75.0%)	40/111 (36.0%)	2.08 (1.12–3.86) ^a^	None
Positive PPD		30/88 (34.1%)	108/313 (34.5%)	0.99 (0.71–1.37) ^a^	None

Abbreviations: BCG, Bacillus Calmette-Guérin; CI, confidence interval; PPD, purified protein derivative; TB, tuberculosis. ^a^ The risk ratio was calculated using the frequencies provided in the publications. ^b^ Relative risk: Adjusted for age, sex, and comorbidities (COPD, asthma, diabetes, hypertension, cancer, renal disease, cardiac disease, and autoimmune disorders). ^c^ Odds ratio: Used imputed datasets to account for incomplete data such as age, sex, race, month of admission, and comorbidities (infection with human immunodeficiency virus, diabetes, hypertension, asthma, malignancy and chronic pulmonary, cardiac and renal diseases).

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
