# Peer review of "Clinical Outcomes of Individuals with COVID-19 and Tuberculosis during the Pre-Vaccination Period of the Pandemic: A Systematic Review"

_jcm, 2022, doi:10.3390/jcm11195656_

Round 1

Reviewer 1 Report

Really interesting and complete work. Limitations are clearly expressed. In my opinion study n.3 pts with  TB family history or workers in contact with TB pts (defined as at risk) should not be considered pts with "TB history" (latent, prior or active) as mantioned in eligibility criteria.

Author Response

Really interesting and complete work. Limitations are clearly expressed.

Point 1: In my opinion study n.3 pts twith TB family history or workers in contact with TB pts (defined as at risk) should not be considered pts with "TB history" (latent, prior or active) as mantioned in eligibility criteria.

Response 1: We have removed the rows for ‘TB family history’, ‘BCG scar’ under the ‘Personal TB exposures’ section, and all the rows in the ‘Occupational TB exposures’ section from Table 3, for both endpoints of hospitalization and radiological pneumonia. We have also removed the column for ‘Personal TB exposures’ and ‘Occupational TB exposures’ since it is no longer needed.

Reviewer 2 Report

This review describes the clinical outcomes of individuals associated with Covid-19 and tuberculosis during the pre-vaccination period of the Covid-19 pandemic. The manuscript is comprehensive and well-written. Although there were relatively few papers that studied the association of clinical outcomes between covid-19 and tuberculosis, authors have done a good job in presenting their perspectives based on available information. As we are aware that there was much misinformation circulating during the early phases of the pandemic regarding the effect of BCG vaccination and Covid-19, and many others; a publication like this is helpful to inform readers about the real scenario.

Authors have concluded that the association between clinical outcomes between Covid-19 and tuberculosis was heterogenous based on different settings and many limitations for comparison. I agree with this observation because tuberculosis is itself a complex disease and the situation of the global scenario of covid-19 adds a different layer of complexity.

I have some minor comments. If the authors address these comments, I recommend publication of this manuscript

1.      Authors have observed that mortality from tuberculosis was associated with increased mortality in high TB burden countries but not in low burden countries such as Canada. I think this point should be further elaborated on. Did the TB cases in high-burden countries were in the advanced stage of disease? Usually in high-burden countries, patients with advanced diseases are the ones who seek medical attention. There may be more clinical and epidemiological features to explain this finding.

2.      In the paragraph dealing with “Mortality, from line number 162, it would be better to specify which specific study by mentioning geographic location or some identifier. For example, in line number 168, “One publication reported…” can be written as “A study from South Africa….”.

3.      In tables 1, 2, and 3, adding the reference number of corresponding studies in a new column will be helpful to follow because in-text referencing is in numerical order.

Author Response

Journal of Clinical Medicine, Manuscript # jcm-1868372

Response to Reviewer 2 Comments

This review describes the clinical outcomes of individuals associated with Covid-19 and tuberculosis during the pre-vaccination period of the Covid-19 pandemic. The manuscript is comprehensive and well-written. Although there were relatively few papers that studied the association of clinical outcomes between covid-19 and tuberculosis, authors have done a good job in presenting their perspectives based on available information. As we are aware that there was much misinformation circulating during the early phases of the pandemic regarding the effect of BCG vaccination and Covid-19, and many others; a publication like this is helpful to inform readers about the real scenario.

Authors have concluded that the association between clinical outcomes between Covid-19 and tuberculosis was heterogenous based on different settings and many limitations for comparison.

I agree with this observation because tuberculosis is itself a complex disease and the situation of the global scenario of covid-19 adds a different layer of complexity.

I have some minor comments. If the authors address these comments, I recommend publication of this manuscript

Point 1: Authors have observed that mortality from tuberculosis was associated with increased mortality in high TB burden countries but not in low burden countries such as Canada. I think this point should be further elaborated on. Did the TB cases in high-burden countries were in the advanced stage of disease? Usually in high-burden countries, patients with advanced diseases are the ones who seek medical attention. There may be more clinical and epidemiological features to explain this finding.

Response 1: We added the following sentence to the Discussion (Lines 310–313):

“On the other hand, in countries with high burden of TB, management becomes more challenging due to high rates of drug-resistant TB,22 increased number of co-morbidities such as HIV-infection,17 and lapse in essential TB services.18,86”

Point 2: In the paragraph dealing with “Mortality, from line number 162, it would be better to specify which specific study by mentioning geographic location or some identifier. For example, in line number 168, “One publication reported…” can be written as “A study from South Africa….”.

Response 2: We thank the reviewer for the suggestion. We have made the suggested modification.

Point 3: In tables 1, 2, and 3, adding the reference number of corresponding studies in a new column will be helpful to follow because in-text referencing is in numerical order.

Response 3: We have added reference numbers in the respective tables.

Reviewer 3 Report

The manuscript written by Tulip Jhaveri et al. represents a comprehensive systematic review regarding the possible relationships between COVID-19 and TB aiming to evaluate the impact of TB on COVID-19 and the impact of COVID-19. Overall, the paper is well presented highlighting the important themes and synthetizing the findings on TB and COVID-19 related with mortality, intertwined adverse effects between these conditions etc.

However, as the authors described, considering the heterogeneity of the analyzed results and the various potential biases revealed by the analyzed papers, I have several minor comments that should be addressed.

1.     Though the authors discussed the contribution of additional factors, including the comorbidities or special populations, it would be advisable that authors to comment two or three sentences on the contribution of these factors in an integrated way (not paper by paper).

2.     As the manuscript well presented, few of the reviewed studies suffer from potential biases that are discussed by the authors. Because is hard to follow these limitations in the manuscript, I suggest that the authors to include a Table synthetizing the common biases and/or methodological issues associated with each analyzed paper.

Author Response

Journal of Clinical Medicine, Manuscript # jcm-1868372

Response to Reviewer 3 Comments

The manuscript written by Tulip Jhaveri et al. represents a comprehensive systematic review regarding the possible relationships between COVID-19 and TB aiming to evaluate the impact of TB on COVID-19 and the impact of COVID-19. Overall, the paper is well presented highlighting the important themes and synthetizing the findings on TB and COVID-19 related with mortality, intertwined adverse effects between these conditions etc.

However, as the authors described, considering the heterogeneity of the analyzed results and the various potential biases revealed by the analyzed papers, I have several minor comments that should be addressed.

Point 1: Though the authors discussed the contribution of additional factors, including the comorbidities or special populations, it would be advisable that authors to comment two or three sentences on the contribution of these factors in an integrated way (not paper by paper).

Response 1: We appreciate the reviewer’s comment. For the discussion of TB as a risk factor for mortality related to COVID-19, lines 340-347 in the Discussion highlight that only a minority of studies reported effect estimates adjusted for potential confounders, which precluded drawing inference about the independent association between TB and mortality. For the discussion of TB as a risk factor for adverse outcomes related to COVID-19, lines 381-384 highlight a similar challenge in drawing inference similar to those referenced above. We propose that this approach is an integrated way to discuss these factors, given that we were not able to use them to draw meaningful inference in both scenarios.

Point 2: As the manuscript well presented, few of the reviewed studies suffer from potential biases that are discussed by the authors. Because is hard to follow these limitations in the manuscript, I suggest that the authors to include a Table synthetizing the common biases and/or methodological issues associated with each analyzed paper.

Response 2: We have added a table in the supplement (Supplementary Table 1) synthesizing common biases and/or methodological issues associated with each analyzed paper, and a reference to this table at the end of the Results section (Line 291).
